# Towards Deeper Understanding of PPR-based Embedding Approaches: A Topological Perspective

## ABSTRACT

Node embedding learns a low-dimensional vectors for each node in the graph. Recent state-of-the-art node embedding approaches take *Personalized PageRank (PPR)* as the proximity measure and factorize the PPR matrix or its adaptation to generate embedding vectors. However, little previous work analyzes what information is encoded by these approaches, and how the information correlates with their superb performance in downstream tasks. In this work, we first show that the state-of-the-art embedding approaches that factorize a PPR-related matrix can be unified into a closed-form framework. Then, we study whether the embeddings generated by such a strategy can be inverted to better recover the graph topology information than random-walk based embeddings. To achieve this, we propose two methods for recovering graph topology via PPR-based embeddings, including the analytical method and the optimization method. Extensive experimental results demonstrate that the embeddings generated by factorizing a PPR-related matrix maintain more topological information, such as common edges and community structures, than that generated by random walks, paving a new way to systematically comprehend why PPR-based node embedding approaches outperform random walk-based alternatives in various downstream tasks. To the best of our knowledge, this is the first work that focuses on the interpretability of PPR-based node embedding approaches.

**ACM Reference Format:**
Anonymous Author(s). 2024. Towards Deeper Understanding of PPR-based Embedding Approaches: A Topological Perspective. In *Proceedings of the ACM Web Conference 2024 (WWW '24), May 13–17, 2023, Singapore.* ACM, New York, NY, USA, 11 pages. https://doi.org/XXXXXXX.XXXXXXX

## 1 INTRODUCTION

Graph data is ubiquitous, ranging from social networks to molecular structures. Given the topological information of the graph, the goal of node embedding is to learn a low-dimensional vector representation for each node. These embeddings play an important role in numerous downstream graph mining tasks, such as link prediction [30, 31, 39], node classification [10, 21, 26], graph reconstruction [36, 37, 41], and recommendation [4, 9, 35, 38].

Node embedding has received significant attention in the last few decades, as it provides valuable insight into effectively leveraging the implicit structural information hidden in the graph. Earlier

attempts [1, 29] can be traced back to 20 years ago, which mainly adopt dimension reduction techniques on graph Laplacian-related matrix to generate node embeddings. These spectral approaches primarily focus on calculating eigenvectors of the graph Laplacian matrix and thus overlook multidimensional connections among users in social networks. To address this limitation, several techniques [27, 28] are proposed to extract latent social dimensions (features) based on graph structures. Subsequently, inspired by the well-known skip-gram model [16, 17], random walk-based embedding approaches [10, 21, 22, 25, 26] demonstrate superior performance in the node classification task. However, these approaches require sampling a large number of multi-hop random walks, posing limitations on their scalability.

Recently, studies on graph neural networks have showcased the effectiveness of using *personalized PageRank (PPR)* to capture crucial graph information [2, 7, 13]. Building upon this observation, several *matrix factorization (MF)*-based node embedding approaches [36, 37, 39] decompose PPR-related matrices, achieving state-of-the-art performances across various graph mining tasks. For example, as shown in Figure 1, the node classification results of the PPR-based embedding approach and one of the state-of-the-art random walk-based embedding approaches, NetMF [22], on two graphs reveal the superiority of the PPR-based node embedding approach. Similar results on other graphs can be found in [36, 37, 39]. Notably, the PPR-based embedding approach outperforms the random walk-based approach by a significant margin. However, limited research study has been conducted to investigate the encoded information within these PPR-based node embeddings or explore how this information facilitates downstream graph mining tasks.

To fill this research gap, in this paper, we aim to provide a comprehensive understanding of state-of-the-art PPR-based node embedding approaches. Our focus lies in investigating the following three fundamental questions, thereby illuminating the inherent information captured by these embedding approaches:

- What specific topological information can be encoded within node embeddings generated by PPR-based MF approaches?
- What is the relationship between spectral approaches and state-of-the-art PPR-based embedding approaches?
- Why do PPR-based embedding approaches consistently outperform random walk-based alternatives?

By addressing these central questions, we aim to provide a significant step towards a deeper understanding of PPR-based MF embedding approaches. A recent work, Deepwalking Backwards [6], explores the information encoded in node embeddings by studying their potential for reconstructing graph topologies. This study focuses on a variant of the classic random walk-based node embedding method, DeepWalk [21, 22], and demonstrates that the node embeddings generated by DeepWalk can approximately recover the community structures in the original graph.

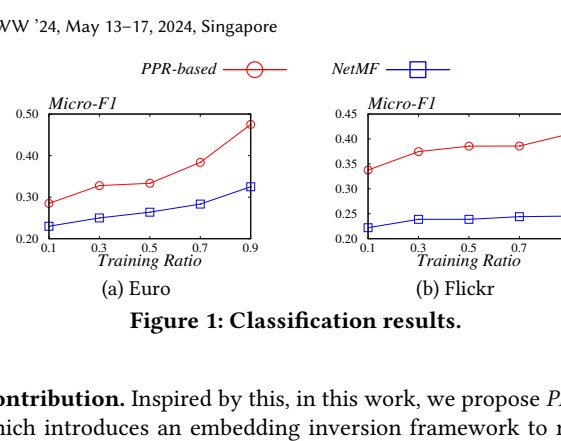

Figure 1: Classification results.

**Contribution.** Inspired by this, in this work, we propose *PPREI*[1], which introduces an embedding inversion framework to reconstruct the original graph using the node embeddings generated by PPR-based MF approach. In particular, we address the aforementioned questions by solving two fundamental problems: the embedding inversion problem and the graph recovery problem. The embedding inversion problem seeks to reconstruct a graph, denoted as $\hat{G}$, from the embeddings of the original graph $G$. The objective is to minimize the disparity between them. On the other hand, the graph recovery problem aims to minimize the dissimilarities in topological structures between $\hat{G}$ and $G$, such as common edges, path lengths, and community structures. The formal definitions are in Section 2.1. In summary, our key findings are as follows.

- We theoretically prove that based on full-rank matrix decomposition, our proposed analytical method can accurately reconstruct the original graph $G$;
- The topological information loss of the graph reconstructed from PPR-based node embeddings is consistently smaller than that of random walk-based node embeddings. Specifically, for PPREI, the relative Frobenius norm error of the adjacency matrix, the relative average path length error, and the relative conductance error of the community in the reconstructed graph are much smaller than Deepwalking Backwards.

To solve the proposed problems, we first show that several state-of-the-art embedding approaches [36, 37, 39] that generate node embeddings by factorizing PPR-related matrices can be unified into a closed-form framework. This framework summarizes the commonalities among representative PPR-based MF embedding approaches. In addition, it can be viewed as a variant of the spectral node embedding approaches, which computes a low-dimensional approximation of a PPR-related graph diffusion matrix.

Subsequently, we focus on this unified framework, which generates node embeddings by computing the low-rank approximation of the PPR-related proximity matrix through singular value decomposition. We introduce two embedding inversion methods: the analytical method and the optimization method. In the analytical method, we establish a connection between spectral analysis and our proposed framework by constructing a linear system. Furthermore, we provide theoretical proof that with full-rank matrix decomposition, we can accurately reconstruct the original graph $G$ using the proposed analytical method, which solves the embedding inversion problem. Our second proposed method is based on directly optimizing an objective function that minimizes the reconstruction error between the proximity matrix $\hat{M}$ calculated on the

reconstructed graph $\hat{G}$ and the proximity matrix $M$ calculated on the original graph $G$.

In our experiments, we compare PPREI with DeepWalking Backwards on the graph recovery task. We evaluate the graph topological properties on 6 real-world graphs, including social networks, flight networks, protein-protein interaction network, and document connection network. Extensive experiments consistently demonstrate that our PPREI outperforms DeepWalking Backwards on all datasets in terms of all evaluation metrics. Our contributions can be summarized as follows.

- We present a closed-form framework that unifies several state-of-the-art PPR-based MF node embedding approaches;
- We present PPREI, a framework that encompasses two PPR-based embedding inversion methods for the graph recovery task;
- Extensive experiments on 6 real-world large graphs demonstrate that PPREI consistently outperforms DeepWalking Backwards in all evaluation metrics, paving a new topological perspective to explain why PPR-based node embedding approaches outperform random walk-based alternatives.

## 2 PRELIMINARIES

### 2.1 Background

**Personalized PageRank.** Let $G = (V, E)$ denote an underlined graph with $n = |V|$ nodes and $m = |E|$ edges, $\boldsymbol{A}$ denote the adjacency matrix, $\boldsymbol{D}$ denote the diagonal degree matrix, and $\boldsymbol{P} = \boldsymbol{D}^{-1}\boldsymbol{A}$ denote the transition matrix. Given a source node $s$, Page et al. [19] first introduce the definition of PPR as follows:

$$\boldsymbol{\pi}_s = (1 - \alpha)\boldsymbol{\pi}_s \cdot \boldsymbol{P} + \alpha \boldsymbol{e}_s, \tag{1}$$

where $\boldsymbol{\pi}_s$ is the PPR vector with respect to source $s$[2], $\alpha$ is the teleport probability and $\boldsymbol{e}_s$ is a one-hot vector with only $\boldsymbol{e}_s(s) = 1$. The PPR vector $\boldsymbol{\pi}(s)$ can be obtained with the Power-Iteration method as shown in [19] by recursively applying Equation 1 until convergence. We can also rewrite Equation 1 into the following matrix form to calculate the PPR matrix $\boldsymbol{\Pi}$:

$$\boldsymbol{\Pi} = \sum_{i=0}^{\infty} \alpha \left( (1 - \alpha)\boldsymbol{P} \right)^i . \tag{2}$$

Although Forward-Push + Monte-Carlo methods [32, 33] are shown to be more efficient for the approximate PPR computation, in this paper, we focus on this matrix form to calculate the proximity matrix due to its mathematically clean definition for analysis. Table 1 lists the notations that are frequently used in this paper.

**Problem formulation.** Following previous work [6], in this paper, we consider two problems, i.e., embedding inversion and graph recovery. Given an embedding algorithm $\mathcal{E}$, the goal of embedding inversion is to generate a graph $\hat{G}$ such that the difference between $\mathcal{E}(G)$ and $\mathcal{E}(\hat{G})$ is negligible.

PROBLEM 1 (EMBEDDING INVERSION). *Let $G = (V, E) \in \mathcal{G}$ be an undirected graph. Given an embedding algorithm $\mathcal{E} : \mathcal{G} \rightarrow \mathbb{R}^{n \times d}$ and embeddings $\mathcal{E}(G)$, the goal of the embedding inversion task is to generate a graph $\hat{G} \in \mathcal{G}$, such that $\mathcal{E}(G) = \mathcal{E}(\hat{G})$ or $\|\mathcal{E}(G) - \mathcal{E}(\hat{G})\|$ is minimized and is negligible for some norm $\| \cdot \|$.*

---

[1]PPR-Based Embedding Invertion.

[2]An entry $\boldsymbol{\pi}_s(v)$ in PPR vector $\boldsymbol{\pi}_s$ indicates the PPR score of $v$ with respect to $s$.

**Table 1: Frequently used notations.**

| Notations | Descriptions |
|---|---|
| $G = (V, E)$ | A graph with node set and edge set |
| $n, m$ | The number of nodes and edges |
| $A, D$ | Adjacent and degree matrix |
| $P, \tilde{L}$ | Transition and normalized Laplacian matrix |
| $\pi(u, v)$ | The PPR value of node $v$ with respect to node $u$ |
| $\alpha, \epsilon$ | Stopping probability and threshold of PPR |
| $K, d$ | Walk length and embedding dimension |
| $M, X, Y$ | Proximity matrix and two embedding matrices |
| $\lambda, v$ | Eigenvalue and eigenvector |

Solving Problem 1 will generate a graph $\hat{G}$, which can be regarded as an approximation of the original graph $G$. Then, a natural question is, to what extent do these two graphs resemble each other in terms of their topological characteristics? Formally, the graph recovery task is defined as follows.

PROBLEM 2 (GRAPH RECOVERY). *Given an embedding algorithm* $\mathcal{E} : \mathcal{G} \rightarrow \mathbb{R}^{n \times d}$, *let* $G, \hat{G} \in \mathcal{G}$ *be two undirected graphs such that* $\mathcal{E}(G) = \mathcal{E}(\hat{G})$ *or* $\|\mathcal{E}(G) - \mathcal{E}(\hat{G})\|$ *is negligible for some norm* $\|\cdot\|$, *the goal of the graph recovery task is to minimize the dissimilarities in topological characteristics between* $G$ *and* $\hat{G}$, *such as common edges, path lengths, and community structures.*

**Remark.** The above two problems focus on analyzing the structural information extracted in the node embeddings. Therefore, the solution to these problems will provide a deeper understanding of node embedding approaches. In this paper, we focus on evaluating the classic random walk method [21, 22] and state-of-the-art PPR-based approaches [36, 37, 39] from a topological perspective. In particular, we will investigate the topological evidence that elucidates *why PPR-based embedding approaches outperform random walk embedding approaches* by solving the above-mentioned problems.

## 2.2 Related Work

Even though embedding approaches that factorize a PPR related matrix have achieved state-of-the-art-performance, to the best of our knowledge, no previous work investigates the reason why such a strategy outperforms random walk embedding approaches. In this section, we first introduce three PPR-based matrix factorization embedding approaches, and then briefly review existing approaches on graph recovery tasks.

**STRAP [37].** Following the basic idea of generating node embeddings from pair-wise PPR, it factorizes an approximation of the PPR matrix calculated on both original graph $G$ and the transpose graph $G^T$. The embedding matrices $X$ and $Y$ are generated as follows:

$$X^T Y = \text{RandomizedSVD}\left(\log\left(\frac{M}{\epsilon}\right), d\right), \quad (2)$$

where $M$ is the proximity matrix and $d$ is the dimension of embedding matrices. Specifically, $M$ is the summation of two approximate PPR matrices, which can be calculated by invoking backward-push [15] algorithm on the original graph $G$ and the transpose graph $G^T$ with threshold $\epsilon$ efficiently.

**NRP [36].** The authors notice that node embeddings derived directly from PPR are sub-optimal. To tackle this issue, they propose a node reweighting algorithm which considers additional node degree information. Specifically, NRP first factorizes a PPR matrix to generate the initial embedding matrices $X$ and $Y$, such that for a pair of nodes $(u, v)$, $X_u^T Y_v \sim \pi(u, v)$, where $\pi(u, v)$ is the PPR value of $v$ with respect to $u$. Then, the node re-weighting algorithm is invoked to preserve a scaled version of $\pi(u, v)$, i.e.:

$$X_u^T Y_v \approx \overrightarrow{w}_u \cdot \pi(u, v) \cdot \overleftarrow{w}_v. \quad (3)$$

**Lemane [39].** Previous MF approaches mainly adopt the same proximity for different tasks, while it is observed that different tasks and datasets may require different proximity. To address this challenge, the authors propose a framework with trainable proximity measures to best suit the datasets and tasks automatically. Instead of factorizing a PPR matrix with fixed $\alpha$, the stopping probability $\alpha_l$ of the random walk at the $l$-th hop is trainable, i.e.:

$$M = \alpha_0 I_n + \sum_{l=1}^{\infty} \alpha_l \cdot \prod_{k=0}^{l-1} (1 - \alpha_k) \cdot P^l. \quad (4)$$

**Graph reconstruction.** Graph reconstruction is the task to recover the graph structures based on embedding vectors. It is treated as a downstream task rather than the optimization objective in most of previous embedding approaches [36, 37, 41]. Another closely related task, graph inversion attack, aims to recover the topological properties of nodes from the perspective of graph privacy. Earlier attempts [6, 8] reconstruct the graph from node embeddings that are generated by DeepWalk or GNN decoder. Link stealing attack [11] proposes to steal links with access to target GNNs, which infers the edges between nodes in the training graph. Another work, GraphMI [42], also aims to recover the links of the original graph by maximizing the classification accuracy of the known node labels. In addition to edge information, Zhang et al. [40] systematically investigates the information leakage of node representations by reconstructing a graph with similar topological properties to the original graph. MNEMON [24] analyzes the implicit graph structural information preserved in node embeddings by model-agnostic attack. A recent study, MCGRA [44], shows that to recover better in attack task, it is essential to extract more multi-aspect information from trained GNN models. Most of above-mentioned attack models are based on GNNs and thus are orthogonal to our work.

## 3 A UNIFIED PPR-BASED EMBEDDING FRAMEWORK

In this section, we first introduce a unified framework for PPR-based MF embedding approaches in Section 3.1. Subsequently, we show that several representative state-of-the-art PPR-based embedding approaches are actually special cases derived from the proposed unified framework. Further details on these specific cases will be provided in Section 3.2.

## 3.1 The Unified Embedding Framework

Graph topology plays an essential role in learning node representations. Recent research studies [18, 36, 39, 43] have demonstrated the effectiveness of PPR values in capturing crucial graph topological information for generating informative node embeddings. Different PPR-based embedding approaches that employ well-designed

 

proximity matrices basically follow similar computation process. This involves factorizing an approximation or adaptation of a PPR-related matrix that preserves weighted similarities between each pair of nodes. Here, we summarize the $K$-hops proximity matrix for these node embedding approaches in the following form:

$$\boldsymbol{M}_K = \max\left\{\mathbf{0}, f\left(\frac{b}{\epsilon K} \cdot \boldsymbol{D}^\beta \sum_{i=k}^{K} \alpha(1-\alpha)^i \boldsymbol{P}^i \boldsymbol{D}^\gamma\right)\right\}, \quad (5)$$

For simplicity, we assume a constant stopping probability $\alpha$ for each step. Notice that our framework can be easily generalized to calculate the proximity matrix with varying stopping probabilities, as we will show in Section 3.2. The hyper-parameter $b$ indicates the bidirectional calculation or helps to alleviate the impact of the threshold $\epsilon$. $f(\cdot)$ represents a non-linear activation function or an identity mapping. Then, we can generate the embedding matrices using singular value decomposition. The $(u, v)$-th entry of the proximity matrix $\boldsymbol{M}_K$ preserves the weighted PPR values of node $v$ with respect to node $u$, which can be expressed as:

$$\boldsymbol{X}_u^T \boldsymbol{Y}_v = \text{RandomizedSVD}(\boldsymbol{M}_K)(u, v)$$
$$\sim g(d_u^\beta) \cdot \pi(u, v) \cdot g(d_v^\gamma),$$

where $g(\cdot)$ denotes a transformation function. Besides, notice that Equation 5 can be rewritten as follows:

$$\boldsymbol{M}_K = \max\left\{\mathbf{0}, f\left(\frac{b}{\epsilon K} \cdot \boldsymbol{D}^\beta \boldsymbol{S} \boldsymbol{D}^\gamma\right)\right\},$$

where $\boldsymbol{S}$ is the graph diffusion matrix with PPR coefficients defined in GDC [14]. Therefore, our proposed unified framework can be viewed as a spectral node embedding approach.

In the subsequent section, we show that the proximity matrices of several representative PPR-based node embedding approaches are, in fact, special cases derived from our proposed unified framework in Equation 5. Consequently, this unified framework establishes the connections among different PPR-based node embedding approaches, enabling us to interpret these embedding approaches from a spectral perspective.

## 3.2 Special Cases

**Interpreting STRAP.** If we set $b = 2K$, $\beta = 0$, $\gamma = 0$, $k = 0$, and $f(x) = \log(x)$, then Equation 5 can be transformed into the following form:

$$\boldsymbol{M}_K = \max\left\{\mathbf{0}, \log\left(\frac{2}{\epsilon} \cdot \sum_{i=0}^{K} \alpha(1-\alpha)^i \boldsymbol{P}^i\right)\right\},$$

which corresponds to the proximity matrix used in STRAP [37]. We summarize this result in the following proposition.

PROPOSITION 1. *Setting $b = 2K$, $\beta = 0$, $\gamma = 0$, $k = 0$, and $f(x) = \log(x)$ in Equation 5 leads to the proximity matrix of STRAP.*

This formulation calculates the PPR values on both the original graph $G$ and the transpose graph $G^T$, such that it preserves both the indegree and outdegree distributions of the given graph. Notice that in the case of undirected graphs, the transpose graph $G^T$ is identical to the original graph $G$, making these two matrices the same. Therefore, we set $b = 2$ to capture such information in the proximity matrix. To incorporate non-linear transformation

operations into the node representations, it adopts log function to get the re-scaling values in $M_K$ before the matrix decomposition operation. The final embedding matrices $\boldsymbol{X}$ and $\boldsymbol{Y}$ can be obtained by decomposing the proximity matrix $\boldsymbol{M}_K$ as follows:

$$\boldsymbol{U}\boldsymbol{\Sigma}\boldsymbol{V}^T = \text{RandomizedSVD}(\boldsymbol{M}_K, d),$$
$$\boldsymbol{X} \leftarrow \boldsymbol{U}\sqrt{\boldsymbol{\Sigma}}, \boldsymbol{Y} \leftarrow \boldsymbol{V}\sqrt{\boldsymbol{\Sigma}}. \quad (6)$$

**Interpreting NRP.** By setting $b = \epsilon K$, $\beta = 0$, $\gamma = 0$, $k = 1$, and $f(x) = x$ in Equation 5, we obtain the following expression:

$$\boldsymbol{M}_K = \sum_{i=1}^{K} \alpha(1-\alpha)^i \boldsymbol{P}^i,$$

which corresponds to ApproxPPR, a truncated version of the PPR matrix used in NRP [36].

Since PPR values for different source nodes are essentially incomparable, the authors further propose a node reweighting technique to mitigate this problem. This technique involves assigning additional weights, $\overrightarrow{w}_u$ and $\overleftarrow{w}_v$, to node $u$ and $v$, respectively. These weights ensure that the $(u, v)$-th entry of the proximity matrix $\boldsymbol{M}_K$ preserves a scaled version of the PPR value $\pi(u, v)$ using approximate node weights:

$$\boldsymbol{M}_K = \boldsymbol{W}_1 \sum_{i=1}^{K} \alpha(1-\alpha)^i \boldsymbol{P}^i \boldsymbol{W}_2.$$

where $\boldsymbol{W}_1$ and $\boldsymbol{W}_2$ are trainable diagonal matrices initialized with the degree matrix $\boldsymbol{D}$. The final embedding matrices $\boldsymbol{X}$ and $\boldsymbol{Y}$ are defined as follows:

$$\boldsymbol{X}_u \boldsymbol{Y}_v^T \approx \boldsymbol{W}_1(u, u) \cdot \pi(u, v) \cdot \boldsymbol{W}_2(v, v).$$

We summarize these findings in the following proposition.

PROPOSITION 2. *By setting $b = \epsilon K$, $\beta = 0$, $\gamma = 0$, $k = 1$, and $f(x) = x$ in Equation 5, the resulting proximity matrix corresponds to ApproxPPR. Moreover, by setting $b = \epsilon K$, $\beta = 1$, $\gamma = 1$, $k = 1$, and $f(x) = x$ in Equation 5, the resulting proximity matrix corresponds to the initial proximity matrix in NRP.*

**Interpreting Lemane.** The authors observe that different tasks and datasets may require different proximity measures. To achieve this, trainable stopping probabilities of random walks are introduced to generate the proximity matrix. By considering trainable parameters $\{\alpha_0, \alpha_1, \cdots, \alpha_K\}$ instead of constant values, our proposed framework can be generalized to compute any truncated proximity matrix. If we set $b = 2K$, $\beta = 0$, $\gamma = 0$, and $f(x) = \log(x)$, Equation 5 can be rewritten as follows:

$$\boldsymbol{M}_K = \max\left\{\mathbf{0}, \log\left(\frac{2}{\epsilon} \cdot \left(\alpha_0 \boldsymbol{I}_n + \sum_{l=1}^{K} \alpha_l \cdot \prod_{k=0}^{l-1} (1-\alpha_k) \cdot \boldsymbol{P}^l\right)\right)\right\},$$

which exactly corresponds to the proximity matrix in Lemane [39]. The following proposition summarizes this result.

PROPOSITION 3. *Setting $b = 2K$, $\beta = 0$, $\gamma = 0$, and $f(x) = \log(x)$ in Equation 5 leads to the proximity matrix of Lemane.*

Similar to STRAP [37], Lemane computes the supervised PPR values on both the original graph $G$ and the transpose graph $G^T$. Therefore, we set $b = 2$ to capture the bidirectional information on

---

**Algorithm 1:** Analytical Method

---

**Input:** Graph $G$, graph volume $vol(G)$, proximity matrix
$\quad\quad\quad\mathbf{M}_K$, stopping probability $\alpha$, threshold $\epsilon$,
$\quad\quad\quad$ propagation step $K$

**Output:** Recovered adjacency matrix $\tilde{\mathbf{A}}$

1 Calculate $\mathbf{M}_\infty$ via Equation 10
2 Calculate $\tilde{\mathbf{L}}$ via Equation 8
3 Calculate $\hat{\mathbf{A}}$ via Equation 9
4 $\tilde{\mathbf{A}} \leftarrow$ Binarize($\hat{\mathbf{A}}$)
5 **return** $\tilde{\mathbf{A}}$

---

undirected graphs. The final embedding matrices $\mathbf{X}$ and $\mathbf{Y}$ can be generated using the same procedure shown in Equation 6.

**Remark.** The proposed unified framework captures the common characteristics of various representative PPR-based embedding approaches, providing a global perspective to interpret these embedding approaches. By employing this framework, the development of novel embedding approaches becomes more straightforward, as it only requires specifying values for the variables in this framework, such as the transformation function $f(\cdot)$, parameters $\epsilon$, $\beta$, $\gamma$, and $k$. In summary, this unified framework simplifies the process of designing interpretable node embedding approaches.

## 4 PPREI

In this section, we present two PPR-based embedding inversion methods: the analytical method and the optimization method. Section 4.1 elaborates on the analytical method, which involves solving a linear system defined within the unified PPR-based embedding framework. Section 4.2 introduces the optimization method, which directly solves an objective function aiming to minimize the differences between the original proximity matrix and the approximate proximity matrix.

### 4.1 Analytical Method

Recap from Section 3.1 that our unified PPR-based node embedding framework is defined as follows:

$$\mathbf{M}_K = \max\left\{\mathbf{0}, f\left(\frac{b}{\epsilon K} \cdot \mathbf{D}^\beta \sum_{i=k}^{K} \alpha(1-\alpha)^i \mathbf{P}^i \mathbf{D}^\gamma\right)\right\}.$$

To simplify the subsequent analysis that establishes a connection between the adjacency matrix and the proximity matrix $\mathbf{M}_K$, we set $b = 1$, $\epsilon = (1-\alpha)/vol(G)$, $\beta = 0$, $\gamma = -1$, $k = 1$, and $f(x) = \log(x)$ in our framework, where $vol(G) = \sum_{i=1}^{n}\sum_{j=1}^{n} \mathbf{A}(i, j)$ is the graph volume. Consequently, we can derive the following expression:

$$\mathbf{M}_K = \log\left(\frac{vol(G)}{K} \cdot \left(\sum_{i=1}^{K} \alpha(1-\alpha)^i \mathbf{P}^i\right)\mathbf{D}^{-1}\right) - \log(1-\alpha), \quad (7)$$

which is consistent with the closed-form expression for the implicit proximity matrix of DeepWalk [22]. Furthermore, Equation 7 can be interpreted as renormalizing the proximity matrix of ApproxPPR in NRP [36] based on the degree distribution $\mathbf{D}^{-1}$.

**Remark.** In addition to NRP [36], similar analysis can also be conducted using STRAP [37] and Lemane [39].

Let $\lambda_i$ and $\boldsymbol{v}_i$ denote the $i$-th eigenvalue and eigenvector of the symmetric transition matrix $\tilde{\mathbf{P}} = \mathbf{D}^{-1/2}\mathbf{A}\mathbf{D}^{-1/2}$. When $i = 1$, we have $\lambda_1 = 1$ and $\boldsymbol{v}_1 = \tilde{\mathbf{D}}^{1/2}\mathbf{e}$, where $\tilde{\mathbf{D}}$ represents the diagonal matrix with entries $\tilde{\mathbf{D}}_{i,i} = d_i/vol(G)$. We can rewrite the expression of $\mathbf{P}^i$ as follows:

$$\mathbf{P}^i = \mathbf{D}^{-1/2}(\sum_{j=1}^{n} \lambda_j^i \boldsymbol{v}_j \boldsymbol{v}_j^T)\mathbf{D}^{1/2}.$$

Therefore, for the proximity matrix $\mathbf{M}_K$ in Equation 7, we have:

$$\mathbf{M}_K = \log\left(\frac{vol(G)}{(1-\alpha)K} \cdot \sum_{i=1}^{K} \alpha(1-\alpha)^i \mathbf{D}^{-1/2}(\sum_{j=1}^{n} \lambda_j^i \boldsymbol{v}_j \boldsymbol{v}_j^T)\mathbf{D}^{-1/2}\right)$$

$$= \log\left(\frac{\alpha \cdot vol(G)\mathbf{D}^{-1/2}}{(1-\alpha)K}\left(\sum_{j=1}^{n}\sum_{i=1}^{K}\left((1-\alpha)\lambda_j\right)^i \boldsymbol{v}_j \boldsymbol{v}_j^T\right)\mathbf{D}^{-1/2}\right).$$

As $|(1-\alpha)\lambda_j| < 1$, when $K \to \infty$, we have $(1-\alpha)\lambda^{K+1} \to 0$. Thus, $\sum_{i=1}^{K}\left((1-\alpha)\lambda_j\right)^i \to \frac{(1-\alpha)\lambda_j}{1-(1-\alpha)\lambda_j}$. Then, inspired by InfiniteWalk [5], we can derive:

$$\mathbf{M}_K = \log\left(\frac{\alpha \cdot vol(G)\mathbf{D}^{-1/2}}{(1-\alpha)K}\left(\sum_{j=1}^{n}\frac{(1-\alpha)\lambda_j}{1-(1-\alpha)\lambda_j}\boldsymbol{v}_j \boldsymbol{v}_j^T\right)\mathbf{D}^{-1/2}\right)$$

$$= \log\left(\mathbf{J} + \frac{\alpha \cdot vol(G)\mathbf{D}^{-1/2}}{(1-\alpha)K}\left(\left(\sum_{j=2}^{n}\frac{(1-\alpha)\lambda_j}{1-(1-\alpha)\lambda_j}\boldsymbol{v}_j \boldsymbol{v}_j^T\right)\mathbf{D}^{-1/2}\right)\right),$$

where $\mathbf{J}$ is the matrix with all entries equal to 1. Since $\lim_{x\to 0} \log(1+x) \to x$, we have:

$$\lim_{K\to\infty} \mathbf{M}_K = \frac{\alpha \cdot vol(G)\mathbf{D}^{-1/2}}{(1-\alpha)K}\left(\sum_{j=2}^{n}\frac{(1-\alpha)\lambda_j}{1-(1-\alpha)\lambda_j}\boldsymbol{v}_j \boldsymbol{v}_j^T\right)\mathbf{D}^{-1/2}$$

$$= \frac{\alpha \cdot vol(G)\mathbf{D}^{-1/2}}{(1-\alpha)K}\left(\sum_{j=2}^{n}\frac{1}{1-(1-\alpha)\lambda_j}\boldsymbol{v}_j \boldsymbol{v}_j^T - \sum_{j=2}^{n}\boldsymbol{v}_j \boldsymbol{v}_j^T\right)\mathbf{D}^{-1/2}.$$

Define $\mathbf{M}_\infty = \lim_{K\to\infty} K \cdot \mathbf{M}_K$. Then we can derive:

$$\mathbf{M}_\infty = \frac{\alpha \cdot vol(G)\mathbf{D}^{-1/2}}{(1-\alpha)}\left(\sum_{j=1}^{n}\frac{1}{1-(1-\alpha)\lambda_j}\boldsymbol{v}_j \boldsymbol{v}_j^T\right)\mathbf{D}^{-1/2}$$

$$- \frac{\alpha \cdot vol(G)\mathbf{D}^{-1/2}}{(1-\alpha)}\left(\frac{1-\alpha}{\alpha}\boldsymbol{v}_1 \boldsymbol{v}_1^T + \sum_{j=1}^{n}\boldsymbol{v}_j \boldsymbol{v}_j^T\right)\mathbf{D}^{-1/2}$$

$$= \frac{\alpha \cdot vol(G)}{(1-\alpha)}\mathbf{D}^{-1/2}\left(\mathbf{Z} - \mathbf{I}\right)\mathbf{D}^{-1/2} - \mathbf{J},$$

where $\mathbf{Z} = \left((1-\alpha)\tilde{\mathbf{L}} + \alpha\mathbf{I}\right)^+$ is a pseudoinverse matrix and $\tilde{\mathbf{L}} = \mathbf{I} - \tilde{\mathbf{P}}$ is the normalized Laplacian matrix.

Consequently, given the proximity matrix $\mathbf{M}_K$ when $K \to \infty$, we can derive the expression of the normalized Laplacian matrix and thus the adjacency matrix:

$$\tilde{\mathbf{L}} = \left(\frac{\mathbf{D}^{1/2}(\mathbf{M}_\infty + \mathbf{J})\mathbf{D}^{1/2}}{\alpha \cdot vol(G)} + \mathbf{I}\right)^+ - \frac{\alpha}{1-\alpha}\mathbf{I}. \quad (8)$$

$$\mathbf{A} = \mathbf{D}^{1/2}(\mathbf{I} - \tilde{\mathbf{L}})\mathbf{D}^{1/2}. \quad (9)$$

**Algorithm 2:** Optimization Method

---

**Input:** Proximity matrix $M_K$, graph volume $vol(G)$,
number of training epoch $p$, iteration number $q$

**Output:** Recovered adjacency matrix $\tilde{A}$

**1** Initialize $\hat{A} \leftarrow \mathbf{0}$

**2 for** $t \in \{1, \cdots, p\}$ **do**

**3**    Initialize $s \leftarrow 0$

**4**    **for** $r \in \{1, \cdots, q\}$ **do**

**5**      $B \leftarrow \sigma(\hat{A} + s)$

**6**      $s \leftarrow s + \frac{vol(G) - \sum_{i,j} B}{\sum_{i,j} (B \otimes (I - B))}$

**7**    $\hat{A} \leftarrow \sigma(\hat{A} + s)$

**8**    Calculate $\hat{M}_K$ via Equation 11

**9**    Calculate $\mathcal{L}$ via Equation 12

**10**    Update $\hat{A}$ to minimize $\mathcal{L}$ using $\frac{\partial \mathcal{L}}{\partial \hat{A}}$

**11** $\hat{A} \leftarrow \text{Binarize}(\hat{A})$

**12 return** $\hat{A}$

---

By calculating Equations 8 and 9, we can exactly recover the original adjacency matrix by solving the above linear system. We summarize the results in the following theorem:

THEOREM 4. *Given an undirected graph $G$ with the full-rank adjacency matrix $A$, the volume of the graph $vol(G)$, and the proximity matrix $M_{K \to \infty}$ with the stopping probability $\alpha$, there exists a linear system such that we can accurately recover the adjacency matrix $A$.*

**Approximation.** In fact, Theorem 4 establishes the connection between the proximity matrix $M_\infty$ and the adjacency matrix $A$. Nevertheless, in real applications, the calculation of the proximity matrix $M_K$ is often infeasible, and thus necessitating an approximation solution. Following a similar analysis of $M_\infty$, when $K$ is sufficiently large, the term

$$\sum_{j=1}^{n} \left((1 - \alpha)\lambda_j\right)^K v_j v_j^T = \left((1 - \alpha)\tilde{P}\right)^K$$

becomes negligible and therefore can be omitted. Based on this analysis, we can derive an approximate expression for $M_K$ with finite $K$ as follows:

$$M_K \approx \log\left(J + \frac{M_\infty}{K}\right)$$

Therefore, given the $K$-hop proximity matrix $M_K$, we can estimate the matrix $M_\infty$ using the following equation:

$$M_\infty \approx K \cdot (\exp(M_K) - J). \tag{10}$$

Consequently, we can recover the adjacency matrix according to Theorem 4. The analytical method is outlined in Algorithm 1. In particular, to generate a binary adjacency matrix $A \in \{0, 1\}^{n \times n}$, following the binarization approach introduced in [6], we set the top-$m$ largest values above the diagonal, as well as their corresponding entries below the diagonal, to 1. This binarization process ensures that the number of edges in the reconstructed graph will be the same as that of the original graph.

**Table 2: Dataset statistics.**

| Name | $n$ | $m$ | $\bar{d}$ | labels |
|------|-----|-----|-----------|--------|
| Euro | 399 | 5,995 | 30.1 | 4 |
| Brazil | 131 | 1,038 | 15.8 | 4 |
| Wiki | 2405 | 17,981 | 15.0 | 17 |
| PPI | 3890 | 76,584 | 39.4 | 50 |
| Flickr | 7575 | 239,738 | 63.3 | 9 |
| BlogCatalog | 10312 | 339,983 | 65.9 | 39 |

## 4.2 Optimization Method

In addition to the analytical method, which returns an approximation of the adjacency matrix based on the given proximity matrix $M_K$, we also propose an optimization method that directly minimizes the gap between the proximity matrices $M_K$ calculated on the original graph $G$ and $\hat{M}_K$ calculated on the recovered graph $\hat{G}$.

Using the embedding matrices $X$ and $Y$, we construct the truncated proximity matrix $M_K = X^T Y$. By treating the entries of the recovered adjacency matrix $\hat{A}$ as trainable parameters, we can optimize these variables using the gradient decent algorithm. We calculate the approximate matrix $\hat{M}_K$ using matrix $\hat{A}$. Specifically, we set $b = K$, $\beta = 0$, $\gamma = 0$, $k = 0$, and $f(x) = \log(x)$ of the proposed unified framework in Equation 5, which is equivalent to calculating the proximity matrix of STRAP on the original graph $G$:

$$\hat{M}_K = \max\left\{\mathbf{0}, \log\left(\frac{1}{\epsilon} \cdot \sum_{i=0}^{K} \alpha(1-\alpha)^i \left(D^{-1}\hat{A}\right)^i\right)\right\}. \tag{11}$$

The objective function $\mathcal{L}$ in our optimization method is defined as:

$$\mathcal{L} = \|\hat{M}_K - M_K\|_F^2. \tag{12}$$

where $\| \cdot \|_F$ denotes the Frobenius norm. The goal of Equation 12 is to minimize the reconstruction error between the original proximity matrix $M_K$ and the approximate proximity matrix $\hat{M}_K$ calculated using the recovered adjacency matrix $\hat{A}$.

Algorithm 2 shows the pseudo-code for the optimization method. It recovers the adjacency matrix by minimizing the objective function introduced in Equation 12. Initially, all entries of $\hat{A}$ are set to zero (Line 1). Then, a shifted logistic function [6] is applied to construct the approximate adjacency matrix using the given graph volume $vol(G)$ (Lines 3-7). Here, $\sigma(\cdot)$ represents the logistic function, $\sum_{i,j}$ calculates the sum of all entries in the matrix, and $\otimes$ denotes the element-wise matrix multiplication. Subsequently, we calculate the approximate matrix $\hat{M}$ via Equation 11, and obtain the objective value of $\mathcal{L}$ through Equation 12. In each epoch, we update $\hat{A}$ by $\frac{\partial \mathcal{L}}{\partial \hat{A}}$ to minimize the objective function. Finally, the binarized approximate adjacency matrix $\tilde{A}$ is returned as the recovered result.

**Remark.** Notice that Algorithm 2 omits the singular value decomposition process due to its instability during the training process and significant computational cost. Instead, we directly construct the truncated proximity matrix $M_K$ from the embedding matrices $X, Y$, and take $M_K$ as input in our optimization method.

## 5 EXPERIMENT

In this section, we first compare two methods of our PPREI on the embedding inversion task. Subsequently, we compare PPREI with

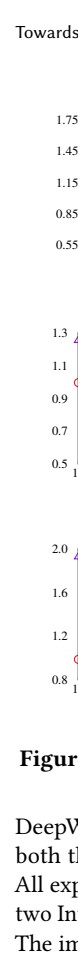

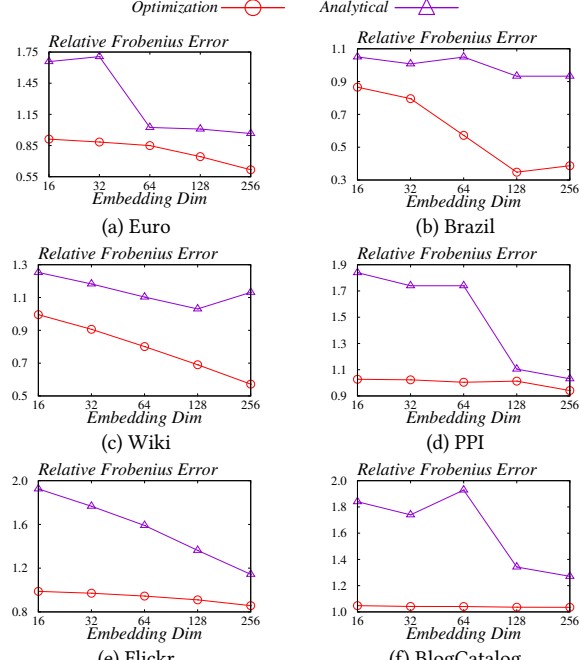

Figure 2: Relative Frobenius error of the adjacency matrix.

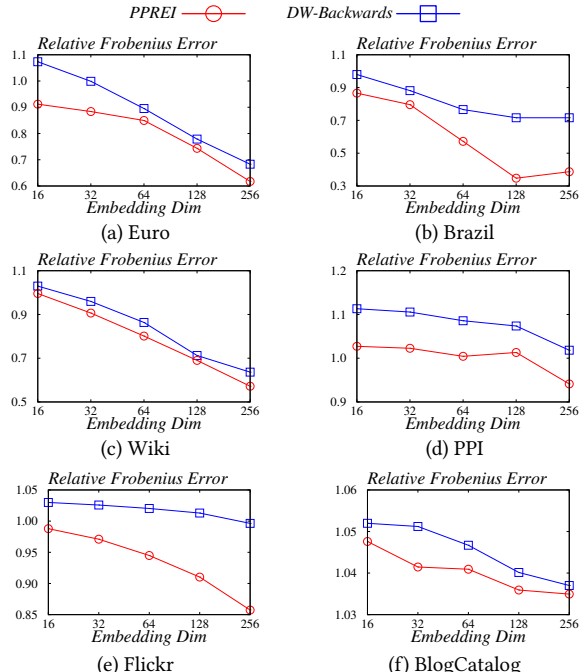

Figure 3: Relative Frobenius error of the adjacency matrix.

DeepWalking Backwards [6], abbreviated as *DW-Backwards*, on both the embedding inversion task and the graph recovery task. All experiments are conducted on a Linux machine equipped with two Intel(R) Xeon(R) CPUs clocked at 2.30GHz and 384GB memory. The implementation of all methods is based on PyTorch [20].

## 5.1 Experimental Settings

**Datasets.** We use 6 real-world datasets that are widely used in recent node embedding studies [21, 23, 30, 34, 36, 39] to evaluate the performance of PPREI and DW-Backwards. Euro and Brazil[3] [23] datasets consist of airport activity records collected from the Statistical of the European Union and the National Civil Aviation Agency, respectively. Wiki[4] [34] dataset is a graph comprising interconnected documents from Wikipedia. PPI[5] dataset [3] represents a subgraph of the protein-protein interaction network for Homo Sapiens. BlogCatalog[6] [27] and Flickr[7] [12] datasets are two social networks where edges indicate friendships/following relationships between users. The statistics of the datasets are shown in Table 2.

**Parameter settings.** For the PPI, BlogCatalog, and Flickr datasets, we set the teleport probability $\alpha = 0.1$, while for the Euro, Brazil, and Wiki datasets, we set $\alpha = 0.7$. In the analytical method, we set the threshold $\epsilon = 10^{-5}$, and in the optimization method, we set $\epsilon = 10^{-7}$. The PPR computation in $M_K$ is conducted with $K = 10$. In the optimization method, the number of training epochs is set to $p = 40$, and the number of iterations for the shifted logistic function is $q = 10$. The dimension of the embedding matrices for each method varies from 16 to 256.

---

[3]https://github.com/leoribeiro/struc2vec/
[4]https://github.com/thunlp/TADW/tree/master/wiki
[5]http://snap.stanford.edu/node2vec/
[6]http://leitang.net/social_dimension.html
[7]https://github.com/mengzaiqiao/CAN/tree/master/data

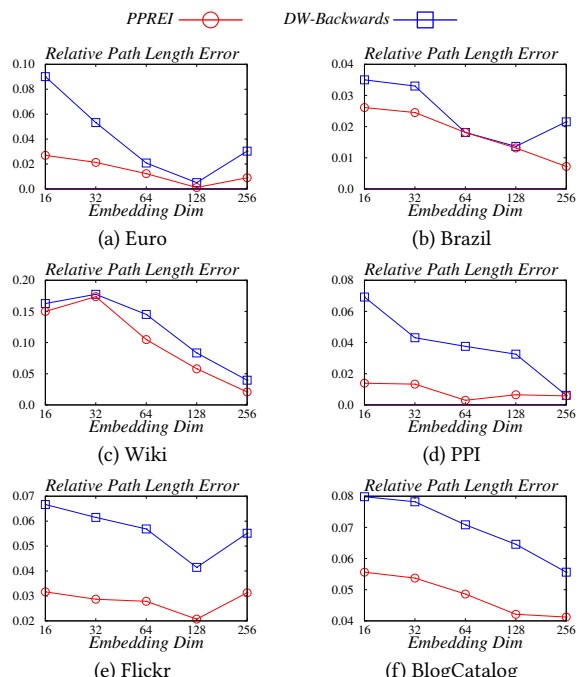

Figure 4: Relative average path length error.

## 5.2 Node Embedding Inversion

In this series of experiments, we evaluate the performance of the proposed analytical method and the optimization method on the node embedding inversion task (defined in Problem 1). Specifically, we evaluate these two methods using the relative Frobenius error, that is, the relative Frobenius norm error between the reconstructed

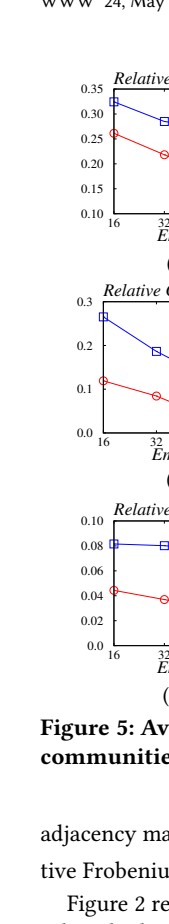

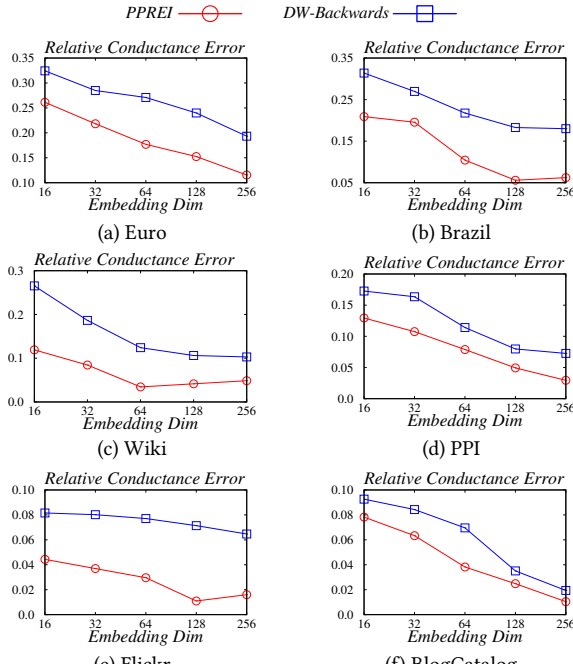

Figure 5: Average relative conductance error of four largest communities.

adjacency matrix $\hat{A}$ and the original adjacency matrix $A$. The relative Frobenius error is computed as $err(A) = \frac{\|A - \hat{A}\|_F}{\|A\|_F}$.

Figure 2 reports the relative Frobenius error of both the analytical method and the optimization method on 6 datasets with varying embedding dimensions $d$. As we can observe, the optimization method consistently outperforms the analytical method by a significant margin. Even though the analytical method can theoretically recover the structure of the graph precisely according to Theorem 4, meeting the conditions stated in the theorem is challenging in practical scenarios. Firstly, the proximity matrix $M_K$ is a truncated approximation of the exact proximity matrix as $K$ is not an infinite number. Secondly, the embedding dimension $d \ll n$, which introduces additional approximation errors to the proximity matrix $M_K$ with a factor of $(1 + \theta)$. On the contrary, the optimization method does not impose constraints on the embedding dimension $d$ or the propagation step $K$.

Based on these observations, we select the optimization method of PPREI for further experimental evaluation.

## 5.3 Graph Recovery

In this set of experiments, we conduct a comprehensive analysis comparing our PPREI with DW-Backwards, aiming to investigate the topological information encoded within PPR-based node embeddings and random walk-based node embeddings. Notice that in addition to the Relative Frobenius norm error $err(A)$ discussed in Section 5.2, we also focus on evaluating the performance on the graph recovery task (defined in Problem 2) by considering two key topological characteristics:

- Relative average path length error $err(l)$: it measures the difference in the average path length between the reconstructed graph

$\hat{G}$ and the original graph $G$. The average path length represents the average distance between all node pairs.

- Average conductance error of the community $err(\phi)$: it measures the difference of the conductance between the reconstructed community and the original community. Given a community $S$, the conductance is defined as $\phi(S) = \frac{\sum_{i \in S, j \in \bar{S}} A(i,j)}{\min\{vol(S), vol(\bar{S})\}}$, where $\bar{S} = V \backslash S$ denotes the complement of community $S$ and $vol(S)$ is sum of the degrees of all nodes in $S$.

Figures 3 to 5 illustrate the relative errors $err(A)$, $err(l)$, and $err(\phi)$ of PPREI and DW-Backwards with varying node embedding dimensions on 6 datasets. Additional experiments analyzing the impact of parameters $\alpha$ and $\epsilon$ can be found in the appendix. Notice that the Euro and Brazil datasets consist of only 4 communities, and therefore, we report the average relative conductance error across the top-4 largest communities on all datasets. The specific relative conductance errors for each community on six datasets are provided in the appendix.

Our observations can be summarized as follows. Firstly, both PPREI and DW-Backwards exhibits a decreasing trend in the relative Frobenius error of the reconstructed adjacency matrix as the embedding dimension increases, as shown in Figure 3. Moreover, PPREI consistently outperforms DW-Backwards on all datasets with different node embedding dimensions. This indicates that PPREI achieves superior performance in preserving the graph topological information. Secondly, the relative path length errors of the graph recovered by PPREI are consistently smaller than those of the graph recovered by DW-Backwards, as shown in Figure 4. This implies that PPR-based node embeddings better preserve the long-range information inherent in the graph. Thirdly, PPREI demonstrates lower average relative conductance errors for the top-4 largest communities compared to DW-Backwards, as shown in Figure 5. This suggests that PPR-based node embeddings better preserve local community information within the graph. Notably, we can observe that when the node embedding dimension $d \geq 128$, PPREI effectively preserves both the average path length and the conductance of communities in the reconstructed graph $\hat{G}$. In summary, these findings demonstrate that PPR-based node embeddings result in significantly less topological information loss compared to random walk-based alternatives. This provides an explanation for the superior performance of PPR-based embedding approaches over random walk-based alternatives from a topological perspective.

## 6 CONCLUSION

In this paper, we provide a comprehensive analysis of PPR-based node embedding approaches. To achieve this, we first introduce a unified PPR-based node embedding framework, which can be considered as a spectral node embedding approach. Subsequently, we show that several representative state-of-the-art PPR-based node embedding approaches can be interpreted as special cases of this framework. Based on this framework, we propose two embedding inversion methods. Extensive experimental results demonstrate that node embeddings generated by PPR-based approaches preserve more accurate graph topological information than that generated by random wark-based approaches. This work enhances our understanding of existing PPR-based node embedding approaches and contributes to advancing the field of graph representation learning.

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

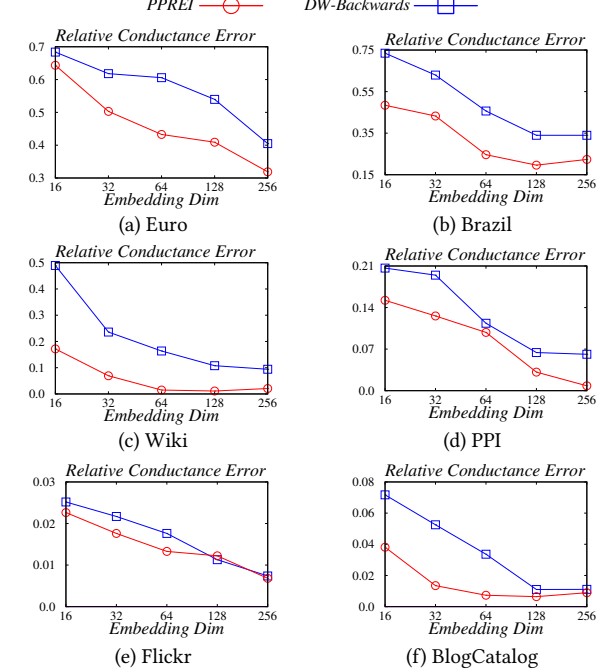

Figure 6: Relative conductance error of the largest community.

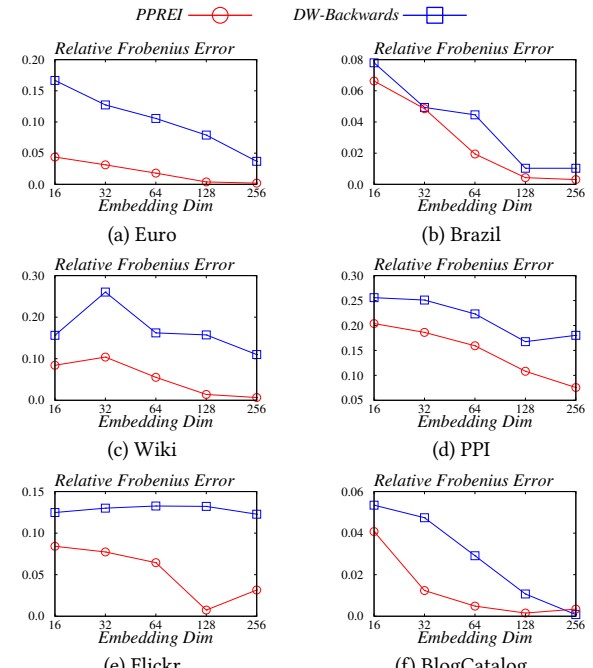

Figure 7: Relative conductance error of the second largest community.

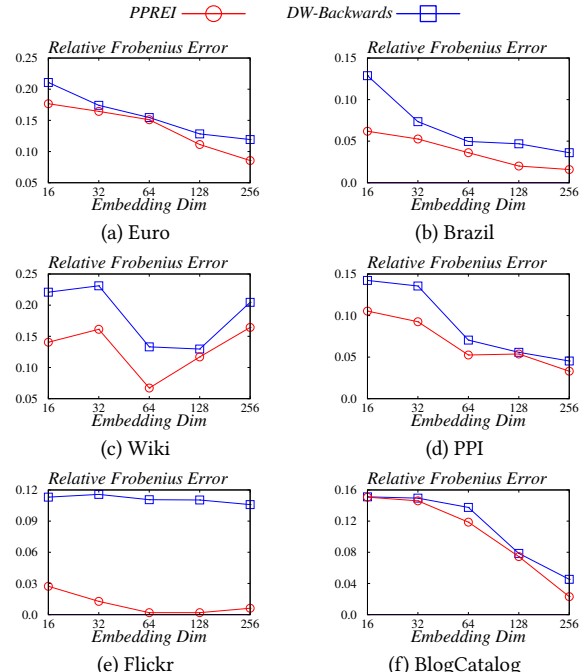

Figure 8: Relative conductance error of the third largest community.

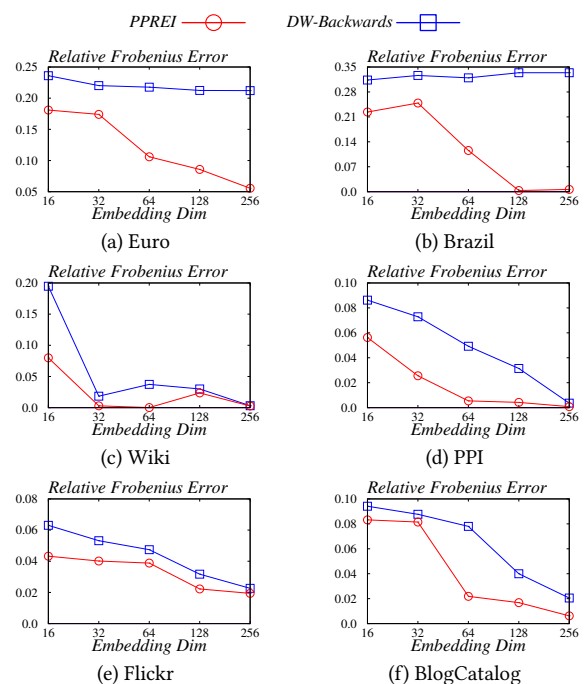

Figure 9: Relative conductance error of the fourth largest community.

## A  EXPERIMENTAL RESULTS

### A.1  Relative Conductance Error

Figures 6 to 9 show the relative conductance errors for top-4 largest communities on 6 datasets. As we can observe, compared to DW

Backwards, PREI exhibits lower relative conductance errors for the top-4 largest communities in most cases. These results again demonstrate that PPR-based node embeddings better preserve local community information within the graph.

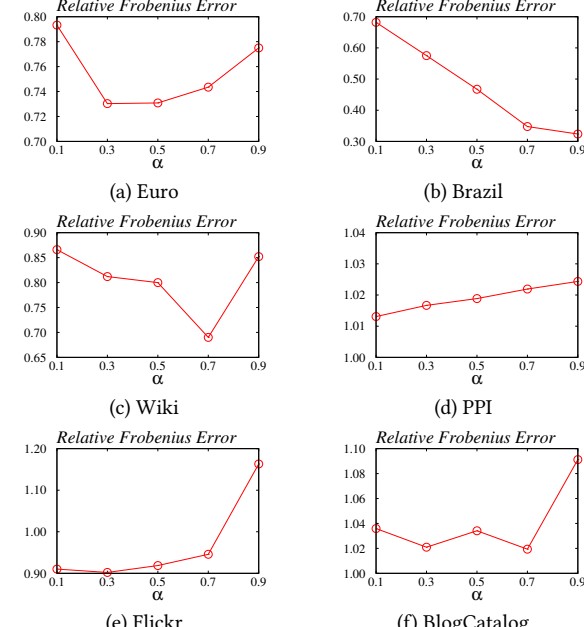

Figure 10: Relative Frobenius error of the adjacency matrix with varying $\alpha$.

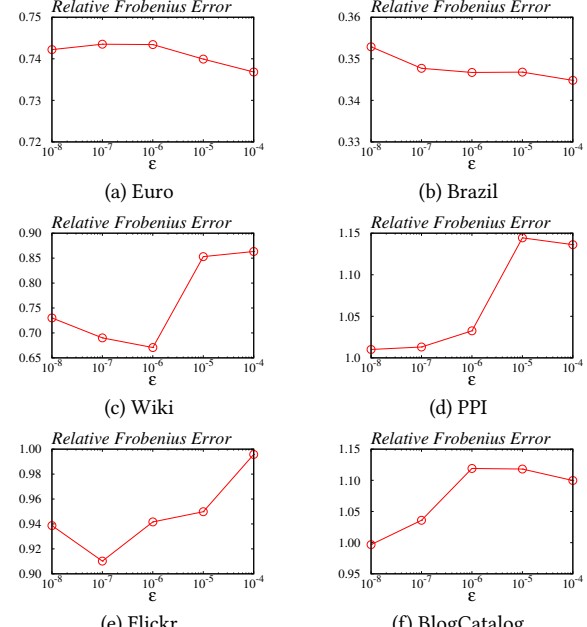

Figure 11: Relative Frobenius error of the adjacency matrix with varying $\epsilon$.

## A.2 Parameter Analysis

In this set of experiments, we investigate the impact of the teleport probability $\alpha$ and the threshold $\epsilon$ on the performance of PPREI by measuring the relative Frobenius norm error $err(A)$ (see Section 5.2). Figures 10 and 11 illustrate the changes in the relative Frobenius norm error on 6 datasets as we vary $\alpha$ from 0.1 to 0.9 and $\epsilon$ from $10^{-8}$ to $10^{-4}$, respectively.

The teleport probability $\alpha$ provides tradeoffs between local and long-range information. As $\alpha$ approaches 0.1, more long-range information will be incorporated into the node embeddings. Conversely, as $\alpha$ approaches 0.9, it will focus more on one-hop neighbors within local communities. As we can observe in Figure 10, on the Euro, Brazil, and Wiki datasets, PPREI achieves competitive results when $\alpha = 0.7$ while on the PPI, Flickr, and BlogCatalog datasets, PPREI exhibits relatively small information loss when $\alpha = 0.1$.

The threshold $\epsilon$ controls the scaling weights of the values in the proximity matrix. Smaller values of $\epsilon$ lead to more discriminative values in the proximity matrix. As we can observe in Figure 11, when $\epsilon = 10^{-7}$, PPREI achieves competitive results and thus $\epsilon$ is set to $10^{-7}$ in our experiments.

