# OpenReview forum: "Towards Deeper Understanding of PPR-based Embedding Approaches: A Topological Perspective"
_ACM.org/TheWebConf/2024/Conference — TheWebConf24 Oral_

### Official Review · Reviewer_f9QW · 2023-11-23

**Novelty:** 4
**Technical Quality:** 5

**Review:**

The paper outlines extensive experiments but could further delve into the practical applications of its findings.A discussion on how these PPR-based node embeddings can be applied in real-world scenarios, such as in social network analysis or bioinformatics, would be beneficial.This would help readers understand the tangible impact of the research. While the paper makes contributions, it would be beneficial to include a more explicit discussion of its limitations.This could involve addressing potential challenges in implementing the PPREI framework in different types of graphs or in more complex scenarios.Additionally, suggesting avenues for future research would be useful.

Pros

Innovative Framework: Introduction of the PPREI framework is a notable innovation.
Comprehensive Theoretical Analysis: Solid theoretical foundation with proofs for the efficacy of proposed methods.
Extensive Empirical Validation: Use of real-world large graphs for testing and comparison with existing methods.
Unified Approach: Unifies various PPR-based MF node embedding approaches into a cohesive framework.
Potential for Wide Application: Implications for a range of graph-related tasks and various fields.

Cons

Limited Visualization Diversity: Reliance mainly on line charts;could benefit from a wider variety of visualization techniques.
Complexity for Non-Experts: Some sections may be challenging for readers not well-versed in the mathematical aspects.
Scalability and Efficiency Considerations: Lacks detailed discussion on the scalability and computational efficiency, which is crucial for large-scale applications.
Open Source Availability: The absence of open-sourced code might limit the ability of other researchers to replicate and build upon the findings.

**Questions:**

Scalability and Computational Efficiency: How does the PPREI framework perform in terms of scalability and computational efficiency, especially when applied to very large graphs? Could you elaborate on any optimizations or challenges faced in this regard?

Generalizability to Different Graph Types: How generalizable is the PPREI framework to different types of graphs, such as dynamic graphs, multi-layer graphs, or graphs with heterogeneous nodes and edges?

Open Sourcing of Implementation Code: Is there a plan to open source the implementation code used in your experiments? How soon can the research community expect its availability, and what kind of support/documentation will accompany it?

Visualizations and Data Presentation: Could you comment on the choice of primarily using line charts for data presentation? Are there other types of visualizations considered or planned for future iterations to enhance data interpretability?

**Ethics Review Description:**

No Ethics problem

**Reviewer Confidence:**

3: The reviewer is confident but not certain that the evaluation is correct

**Scope:**

3: The work is somewhat relevant to the Web and to the track, and is of narrow interest to a sub-community

---

### Official Review · Reviewer_Du3i · 2023-11-24

**Novelty:** 4
**Technical Quality:** 3

**Review:**

In this work, the authors first show that the state-of-the-art embedding approaches that factorize a PPR-related matrix can be unified into a closed-form framework. Then, they study whether the embeddings generated by such a strategy can be inverted to better recover the graph topology information than random-walk based embeddings. To achieve this, they propose two methods for recovering graph topology via PPR-based embeddings, including the analytical method and the optimization method. Extensive experimental results demonstrate that the embeddings generated by factorizing a PPR-related matrix maintain more topological information. The main innovation of this paper is providing a  novel insight excavating the capacity of PPR and giving a somewhat mature framework to solve downstream tasks.

The concrete cons are listed here:

1. The paper lacks a more reasonable explanation for motivation, and it is unreasonable to explain everything with prior experimental results. Specifically, a more detailed explanation about the motivation of PPR-based embedding will be of great value.
2. The composition of the architecture is not clearly described. Apparently, there is not even a schematic diagram for reference, which is shocking me a lot. Personally speaking, I would rather understand the model architecture with diagrams than read your lengthy text description.
3. The paper gives me a noel insight and seems reasonable, I must say. However, there is no code providing in supplementary materials and the proposed model cannot be further verified.
4. The experiment is insufficient and the overall workload is limited. The setting of downstream tasks seems to need an explanation about the rationality: only graph-level experiments are carried out, and the more commonly used node-level and edge-level tasks are not discussed at all, as if the model is strongly coupled with downstream tasks. As we all know, this non-uncoupled model seems to be disadvantageous for the community to generalize it, which may require the author to supplement or explain the necessity of the original setting. Moreover, there was no ablation experiment.

**Questions:**

See above cons.

**Reviewer Confidence:**

3: The reviewer is confident but not certain that the evaluation is correct

**Scope:**

3: The work is somewhat relevant to the Web and to the track, and is of narrow interest to a sub-community

---

### Official Review · Reviewer_t29d · 2023-11-24

**Novelty:** 5
**Technical Quality:** 5

**Review:**

This paper studies the interpretability of employing Personalized PageRank (PPR) for the purpose of generating node embedding vectors. This research presents a comprehensive framework that aims to integrate the latest PPR-based embedding methods, as they may be seen as a modified version of the spectral node embedding method. Following this paradigm, two methods for graph topology recovery are provided, utilizing PPR-based embedding. These methods consist of the analytical method and the optimization method. Ultimately, comprehensive experimental findings demonstrate that the embeddings produced by the factorization of a PPR-based matrix possess a greater amount of topological information compared to the embeddings formed through random walking.

pros
1. This paper is a pioneering work in the field of interpretability research on PPR-based node embedding methods.
2. Starting from the spectral node embedding method, this paper unifies a variety of PPR-based node embedding methods through different parameter settings. This approach simplifies the process of designing an interpretable node embedding method.
3. The proof of the theorem in this paper is relatively rigorous. The solution process of the linear system under the embedding framework is very detailed and clear, which provides a strong argument for this method.

cons
1. Sufficient comparative experiments are carried out in the experiment, which proves the feasibility of the method to a certain extent. However, there are problems of small scale and homogeneity in the selection of datasets.
2. Two PPR-based embedding inversion methods are proposed in this paper, and the optimization method is proved to be better than the analysis method through experiments, but there is a lack of introduction to the application scenarios of the two methods, or the optimization method can be used directly.
3. The paper mentions that the use of frameworks simplifies the process of designing explainable node embedding methods, so does it lead to an improvement in the results, which has not been verified by experiments.
4. The source code is not released.

**Questions:**

1. Are the two PPR-based embedding inversion methods proposed in this paper also applicable to large data scales?
2. Does the use of the framework affect the experimental results of PPR-based embedding inversion or only serve to simplify the process?
3. According to the experimental results, the optimization-based embedding inversion method is much better than the analysis-based method, so can the former completely replace the latter?

**Reviewer Confidence:**

3: The reviewer is confident but not certain that the evaluation is correct

**Scope:**

4: The work is relevant to the Web and to the track, and is of broad interest to the community

---

### Official Review · Reviewer_QNgw · 2023-11-26

**Novelty:** 4
**Technical Quality:** 6

**Review:**

This paper is focused on the problem of reconstructing a graph based on personalized pagerank (PPR) embeddings. A unified view of different PPR embeddings is provided together with two reconstruction algorithms, one analytical and one based on optimization. The experiments, based on six real datasets show that the proposed reconstruction algorithms outperform their counterparts based on deepwalk embeddings in terms of multiple reconstruction metrics.

Strengths:

+The paper is well-written and easy to follow

+The paper shows that PPR embeddings are better at graph reconstruction

+The paper seems technically solid

Weaknesses:

-The contributions of the paper are limited

-Evaluation metrics are not comprehensive

-Deepwalk embeddings are not necessarily good for graph reconstruction

Detailed comments:

I overall enjoyed reading this paper even though it has important limitations that I will detail here.

*Contributions: The proofs and algorithms proposed in the paper are quite similar to the ones in [6]. The differences between the papers, except for the embedding matrix, should be more clear.

*Evaluation: The experiments should include additional results, such as visualizations or the real and reconstructed graphs and other graph statistics such as degree distribution, clustering coefficient, diameter, etc.

*Deepwalk: It has been shown that embeddings based on deepwalk embeddings are not effective at link prediction, see reference below for stronger baselines:
https://openreview.net/pdf?id=EoDpq18R30

**Questions:**

1) What are the specific differences between this paper and [6]?

2) What do the predicted graphs look like (visualizations, statistics, etc)?

3) How do PPR-based reconstruction methods compare with strong baselines (see comments above)?

**Reviewer Confidence:**

4: The reviewer is certain that the evaluation is correct and very familiar with the relevant literature

**Scope:**

4: The work is relevant to the Web and to the track, and is of broad interest to the community

---

### Official Review · Reviewer_DdE7 · 2023-11-28

**Novelty:** 4
**Technical Quality:** 4

**Review:**

This paper provides a comprehensive understanding of state-of-the-art PPR-based node embedding approaches. The authors analyzed the PPR-based MF embedding approaches, and theoretically prove that based on full-rank matrix decomposition, their analytical method can accurately reconstruct the original graph, and the topological information loss of the graph reconstructed from PPR-based node embeddings is consistently smaller than that of random walk-based node embeddings. Their experiments on 6 real-world large graphs demonstrate that the proposed method can consistently outperform DeepWalking Backwards in all evaluation metrics. The writing of this paper is good, and the theoretical analysis is comprehensive. The authors also conduct experiments on 6 different datasets, which makes the experiment results more convincing.

**Questions:**

This paper discusses the relationship between Personalized PageRank based embedding approaches and the random walk based embedding approaches. But is it possible to compare PPR-based embedding approaches with random walk with restart-based approaches? The formula in Equation (1) is very similar to the formula of RWR.

Many works learn the node embedding with graph neural networks. Is it possible to discuss the relationship between PPR-based embedding approaches and GNN based methods?

**Reviewer Confidence:**

1: The reviewer's evaluation is an educated guess

**Scope:**

3: The work is somewhat relevant to the Web and to the track, and is of narrow interest to a sub-community

---

### Decision · Program_Chairs · 2024-01-22

**Decision:**

Accept (Oral)

**Comment:**

The paper makes a solid contribution to the study of PPR-based embeddings, unifying some of them under a common framework and studying how they recover graph topology.

 The unifying scheme does not seem to cover VERSE [30], yet an effort to express it via matrix factorization, and an association to STRAP, which is coverd in this paper, is made in:
 FREDE: Anytime Graph Embeddings. Proc. VLDB Endow. 14(6): 1102-1110 (2021)